# Behavior of Rectangular-Sectional Steel Tubular Columns Filled with High-Strength Steel Fiber Reinforced Concrete Under Axial Compression

**DOI:** 10.3390/ma12172716

**Published:** 2019-08-24

**Authors:** Shiming Liu, Xinxin Ding, Xiaoke Li, Yongjian Liu, Shunbo Zhao

**Affiliations:** 1School of Civil Engineering and Communication, North China University of Water Resources and Electric Power, Zhengzhou 450045, China; 2Key Laboratory for Bridge Detection and Reinforcement Technology of Ministry of Communications, Chang’an University, Xi’an 710064, China; 3International Joint Research Lab for Eco-building Materials and Engineering of Henan, North China University of Water Resources and Electric Power, Zhengzhou 450045, China

**Keywords:** rectangular-sectional concrete-filled steel tube column, steel fiber reinforced concrete, perfobond rib, steel plate rib, axial compression, bearing capacity, ductility, energy dissipation capacity

## Abstract

This paper studies the effect of high-strength steel fiber reinforced concrete (SFRC) on the axial compression behavior of rectangular-sectional SFRC-filled steel tube columns. The purpose is to improve the integrated bearing capacity of these composite columns. Nine rectangular-sectional SFRC-filled steel tube columns and one normal concrete-filled steel tube column were designed and tested under axial loading to failure. The compressive strength of concrete, the volume fraction of steel fiber, the type of internal longitudinal stiffener and the spacing of circular holes in perfobond rib were considered as the main parameters. The failure modes, axial load-deformation curves, energy dissipation capacity, axial bearing capacity, and ductility index are presented. The results identified that steel fiber delayed the local buckling of steel tube and increased the ductility and energy dissipation capacity of the columns when the volume fraction of steel fiber was not less than 0.8%. The longitudinal internal stiffening ribs and their type changed the failure modes of the local buckling of steel tube, and perfobond ribs increased the ductility and energy dissipation capacity to some degree. The compressive strength of SFRC failed to change the failure modes, but had a significant impact on the energy dissipation capacity, bearing capacity, and ductility. The predictive formulas for the bearing capacity and ductility index of rectangular-sectional SFRC-filled steel tube columns are proposed to be used in engineering practice.

## 1. Introduction

The steel-concrete composite column, also called as concrete-filled steel tube (CFST) column, has been widely applied in buildings and bridges as the structural members such as column, arch rib, pylon, abutment, and pier. By January 2015, there were 413 CFST arch bridges with a span no less than 50 m in China. This takes the advantages of high compressive strength of in-filled concrete and rational ductility of steel tube, and overcomes the inward buckling of the steel tube due to the support of in-filled concrete [1,2,3,4,5]. In addition, as the steel tube acts not only as the stay-in-place formwork during the casting of in-filled concrete but also as the external reinforcement which diminishes a large number of internal reinforcing bars, both the engineering cost and period of construction are greatly reduced [6].

Owing to easy connection and working reliably together with other structural members such as beam, wall, and slab when compared with circular-sectional CFST column, rectangular-sectional and square-sectional CFST columns have attracted more and more interests around the world [7,8]. Compared with square-sectional or circular-sectional CFST columns, rectangular-sectional CFST column has the unequal bending stiffness along different axes. This is ideally suitable to fit the mechanical behaviors of the members including arch rib, pylon, abutment, and pier of bridges, and other structural members under loading actions varied greatly in longitudinal direction to transverse direction. However, the uneven and relatively weak confinement along the longer side of steel tube to core concrete are induced, and the load-carrying capacity and the ductility under axial load are reduced. Therefore, how to improve the interfacial bond property between the rectangular-sectional steel tube and the core concrete have been a key problem in engineering practices.

One method is to use the high-strength concrete as the core concrete and improve the interfacial bond property. This also benefits to the increase of load-carrying capacity and the reduction of cross-sectional dimensions [9,10,11,12,13,14,15], however with the high-strength concrete accompanies a problematic shortcoming of the inherent brittleness and large volume shrinkage [16,17]. Another method is by means of the longitudinal stiffening rib which is welded on the inner surfaces of the rectangular-sectional steel tube [18,19,20,21,22,23]. The stiffening rib can serve as the internal reinforcing bar and effectively improves the bond between the steel tube and the core concrete. Typical stiffening rib includes the steel plate rib and the perfobond rib made of the steel plate with equal spacing circular holes. Owing to the advantage of easily installed and good ductility under compression, the CFST column stiffened with perfobond ribs has been widely accepted by the construction companies [24]. The stiffening rib can also increase the bearing capacity of the CFST column and delay the buckling of steel tube, however, it increases the construction cost to some degree.

To improve the ductility of CFST column, high-strength or heavy plate is widely applied [25,26], which inevitably causes the increase of engineering cost and welding difficulty during the construction. Another way is the use of rubberized concrete as the core concrete. The rubberized concrete is made of the normal aggregate partially replaced by recycled tire rubber aggregate. Rubberized concrete can significantly improve the ductility [27] and decrease the engineering cost and natural frequency of a structural member [28,29]. The bending and seismic performances of circular-sectional, square-sectional, and rectangular-sectional CFST columns with rubberized concrete have become an active research topic [30,31]. Compared with normal concrete, the structural application of rubberized concrete is limited due to the decreased compressive strength. Meanwhile, with the development of high-flowing and self-compacting steel fiber reinforced concrete (SFRC) [32,33], perfect distribution pattern of steel fiber in high-strength concrete comes into reality [34,35,36,37,38], and high interfacial bond performance between the steel tube and the in-filled SFRC can be obtained [39,40,41]. This provides a new way to enhance the ductility of CFST columns. In this respect, the compressive behavior of circular-sectional and square-sectional SFRCFST columns have become one of the research hotspots in recent years, and the major factors consisted of the cubic compressive strength of SFRC ranging from 23.5 to 115 MPa [42,43], the type of steel fiber including hooked-end and crimped [44,45], and the volume fraction of steel fiber varying within 0.2–2% [46,47]. To minimize the increased cost due to the addition of steel fibers, the design program was developed and applied into practice for the circular-sectional SFRCFST column [48]. With the cubic compressive strengths of SFRC from 32.4 to 57.8 MPa and the volume fraction of steel fiber from 0.75% to 1.25%, the lateral cyclic loading behavior of circular-sectional SFRCFST columns were investigated [49,50,51,52]. The results pointed out that the ductility, compressive rigidity, and energy absorption capacity of the columns could be improved by the presence of steel fibers, and the minimum volume fraction of steel fiber was 0.75% to get the sufficient ductility.

Based on above reviews, both the longitudinal stiffening rib and SFRC can be applied to enhancing the behavior of CFST columns under axial compression. This constitutes the focus of the research topic of this paper. Ten rectangular-sectional columns were experimentally studied under axial compression. The SFRC with different compressive strengths compared with normal concrete, the volume fraction of steel fiber, the stiffening rib type, and the spacing of circular holes in perfobond rib were considered as the main parameters. The failure modes, axial bearing capacity, and axial load-deformation curves are presented. The simplified predictive formulas for the bearing capacity and ductility index are proposed.

## 2. Experimental Works

### 2.1. Preparation of Concrete

Considering the different compressive strength and the variation of volume fraction of steel fiber for the core SFRC of tested SFRCFST columns, six concretes were designed with expected cubic compressive strength of 60, 70, and 80 MPa at 28 days respectively. Their mix proportions are presented in Table 1, where the identifier starting with “C” and “CF” refers to normal concrete and SFRC, and the following numbers denote the expected cubic compressive strength and the volume fraction of steel fiber. The raw materials included ordinary silicate cement, slag powder, river sand, coarse aggregate with a maximum particle size of 20 mm, water reducer, and tap water. Hooked-end steel fiber was produced by Shanghai Harex Steel Fiber Technology Co., Ltd (Sahnghai, China) with the length of 30 mm and diameter of 0.75 mm; the aspect ratio was 40 and the tensile strength was larger than 600 MPa. The slump of mixtures was greater than 550 mm.

### 2.2. Design and Preparation of Specimens

Ten specimens of rectangular-sectional CFST columns were designed and prepared. Eight of them were stiffened with perfobond ribs having different spacing of circular holes and in-filled SFRC with a varying volume fraction of steel fiber, one was stiffened with steel plate ribs and the other one was no rib. The cross-sectional dimension was 360 mm × 188 mm, and the height was 900 mm. The slenderness ratios of the specimens, i.e., the effective length divided by the radius of gyration, were 8.7 and 16.6 along the two main principal axes. The specimens belong to short columns in axial compression with slenderness less than 17.5 [53]. Details of different stiffeners are presented in Figure 1.

For each specimen, six triangular stiffeners were arranged at the bottom steel end plate. The thickness of the rectangular-sectional steel tube and longitudinal internal stiffening rib, the triangular stiffener and the end-steel plate were 4, 8, and 20 mm, respectively. The yield strength, ultimate strength, and elastic modulus of the steel were 360 MPa, 576 MPa, and 206 GPa, respectively.

Table 2 presents the details of mechanical properties and stiffener type of the specimens, where the specimen ID with A, B, and C donated the stiffener type of perfobond rib, steel plate rib, and no rib, respectively; followed by the hole spacing of perfobond rib, the strength grade of core concrete and the volume fraction of steel fiber. The mix proportions of core concrete (Table 1) for the specimens can be identified from the third part of the ID, as presented in Table 2. The cubic compressive strength *f*_cu_ and the axial compressive strength *f*_c_ of core concrete were tested by the standard cubes with dimension of 150 mm and the standard prisms of 150 × 150 × 300 mm as per China codes [54,55]. Six cubes and six prisms were fabricated for a mixture, three of them as a group were tested on the testing machine. They were cured at the same condition as the column specimens.

In order to estimate the confinement of rectangular-sectional steel tube to core concrete, the confinement coefficient *θ* is defined as [11].
(1)$$\theta =\frac{f_{yt}A_{st}}{f_{c}A_{c}},$$
where *f_yt_* is the yield strength of the steel tube (MPa); *A_st_* and *A_c_* are the cross-sectional area of the steel tube and the core concrete (mm^2^), respectively.

All the specimens were fabricated as per China codes for SFRC and welding of steel structures [54,56]. A single-horizontal-shaft forced mixer was used to mix the fresh normal concrete and SFRC. An automatic arc machine was used to weld the steel plate. A mobile ultrasonoscope was used to do the quality inspection of welding seams.

The rectangular-sectional steel tubes and stiffeners were manufactured with steel plates. The circular holes were bored for perfobond rib. Sand blasting was taken for the stiffeners. The rectangular-sectional steel tubes were formed with stiffeners and welded with the bottom end-steel plate together. After that, the core concrete was pumped into the rectangular-sectional steel tube in layers, and vibrated by the electrical vibrator. Then the specimens were covered by using straw mattress and cured by spraying water for 28 days at room temperature.

After the top-surface being roughed with a steel wire brush, a thin layer of high-quality high-strength cement paste was overlaid to flush the top-surface of core concrete with the rectangular-sectional steel tube, and then the top end-steel plate was welded on to the steel tube.

### 2.3. Test Method

As exhibited in Figure 2, the specimens were conducted by using a 1000-ton universal testing machine which has a maximum load of 10,000 kN. To detect the local deformation, 30 × 30 mm black grids were plotted on the surface of specimens before testing. The axial load was step-by-step applied with force control at a speed of 10 kN/s when the force was lower than 70% of the estimated ultimate load, and then shifted with displacement control with an increment of 0.8 mm/min until failure. A load step about 7.5% of the estimated ultimate load was adopted, and each step was maintained for no less than 2 min to get the stable data [57].

Four linear variable differential transducers (LVDTs) were symmetrically arranged near the corners of the specimens, and the mean value of four LVDTs was defined as the axial deformation. Twelve strain gauges were installed in middle of the specimens to measure the axial and lateral strains of the steel tube. All the experimental data were synchronously recorded by the data collecting instrument.

## 3. Test Results and Discussions

### 3.1. Failure Modes

An abnormal sound was made when the load was up to 50–60% of the maximum load, and then the sound died away. The specimen without stiffener was prone to have drum shape failure mode, while the stiffened specimens presented the local buckling of the steel tubes along their long sides. The crushed concrete core was obviously found after the external steel tube was removed.

The typical failure modes of A-60-CF70/1.2, B-0-CF80/1.2, and C-0-CF80/1.2 are shown in Figure 3, and the local buckling of steel tubes appeared along the long sides where the distance from the upper end was 390, 255, and 170 mm, respectively. For the columns stiffened with perfobond ribs, the local buckling appeared initially at the position of the holes where cross-sectional weakness was significant. For the stiffened columns, the local buckling of the steel tubes occurred outward on their longer sides, and the failure appearances were significantly affected by the stiffeners (Figure 4a). In the case of the unstiffened column, all the steel tubes buckled outward, with the nodes at the corners of the column (Figure 4b).

To study the damage degree of the core concrete, the steel tube was cut off, and the failure appearances of core concrete can be seen in Figure 5. The core concretes corresponding to the steel tube buckling were crushed along their long dimensions and not influenced by the stiffeners which were different from the steel tubes, and some cracks appeared on the surface of the core concretes. It was noted that steel fibers were entirely pulled out without breaking, which indicated that the tensile strength being larger than 600 MPa was fit for the CFST columns under axial load.

### 3.2. Axial Load-Deformation Curves

The axial load-deformation curves of the test columns affected by the hole spacing, the volume fraction of steel fiber, and the stiffener type are exhibited in Figure 6, Figure 7 and Figure 8, respectively. As expected, these curves consist of three segments presented as linear ascending, softening, and descending. The slopes of ascending curve and descending curve were used to evaluate the compressive rigidity and post-yield axial ductility of the column under axial load, respectively. For the convenience of quantitative analysis, the related values of slopes are defined as the axial shortening of specimens less than 1.5 mm and more than 4 mm, respectively. The compressive rigidity increases with the concrete strength. However, the post-yield axial ductility decreases with the concrete strength.

In the condition of the same volume fraction of steel fiber and the equivalent compressive strength of core concrete for the SFRCFST columns, as presented in Figure 6, the slopes of ascending curve of A-120-CF80/1.2, A-90-CF80/1.2 and A-60-CF80/1.2 are 3500.0, 3265.9, and 3500.3 kN/mm, and the slops of descending curve are −446.4, −419.5, and −415.5 kN/mm. Compared to A-60-CF80/1.2 with the hole spacing as two times the diameter, A-120-CF80/1.2 with the hole spacing as four times the diameter had an equivalent compressive rigidity and 6.7% decrement of post-yield axial ductility, and A-90-CF80/1.2 with the hole spacing as three times the diameter had 7.2% decrement of compressive rigidity and 1.0% decrement of post-yield axial ductility. Therefore, for the SFRCFST columns stiffened with perfobond ribs, the hole spacing can be taken as two times the diameter to get a larger compressive rigidity and post-yield axial ductility.

In the condition of the columns with the same perfobond ribs, Figure 7 exhibits that due to the combination of the changes of compressive strength of SFRC with a varying volume fraction of steel fiber, the effect of steel fiber on the compressive rigidity of SFRCFST columns did not clearly appear. However, the slope of the descending segment of curves became gentle with the increasing volume fraction of steel fiber. This means that the post-yield axial ductility of SFRCFST columns increased with the volume fraction of steel fiber, especially when the axial shortening was over 4 mm. This is due to the confinement of steel fibers to the transversal deformation of core concrete for SFRCFST columns. The confinement increased with volume fraction of steel fiber, and postponed the failure after the appearance of cracks accompanied with the local buckling of steel tube.

In the condition of the same volume fraction of steel fiber for SFRC of the SFRCFST columns, as exhibited in Figure 8, the compressive rigidity of the columns generally increased with the compressive strength of core SFRC. However, the best post-yield axial ductility took place on the SFRCFST column stiffened with perfobond ribs. The larger post-yield axial ductility of the column C-0-CF80/1.2 was due to the higher compressive strength of SFRC. This indicates that the stiffener type had a higher effect on the post-yield axial ductility of SFRCFST columns.

### 3.3. Energy Dissipation Capacity

The energy dissipation capacity *E* is defined as the area of axial load-deformation curves till 85% of ultimate force *N*_u_ in the descending segment. The computed values of the specimens are presented in Table 3. Compared to the CFST column A-60-C70/0, the energy dissipation capacity of SFRCFST columns had the increment of 42.3%, 31.3–43.1% and 51.3% respectively with the volume fractions of steel fiber as 0.8%, 1.2% and 1.6%. This declares that the addition of steel fibers significantly increases the energy dissipation capacity of SFRCFST columns. In this aspect, the greater transversal deformation capacity of core SFRC was given out with the confinement of steel fibers until the pull-out owing to the successive expansion.

Compared to the column C-0-CF80/1.2 without rib, the energy dissipation capacity of B-0-CF80/1.2 stiffened with steel plate ribs had a little decrease of 2.8% due to the lower concrete strength, but the equivalent columns stiffened with perfobond ribs increased about 2.7–9.3%. This comes down to the dowel action of core concrete across the holes of perfobond ribs which increased the energy dissipation capacity. The columns stiffened with perfobond ribs had a larger energy dissipation capacity than that stiffened with steel plate ribs, and the effect of the hole spacing of perfobond ribs was not significant.

In the condition of the same other parameters, the energy dissipation capacity of the SFRCFST columns increased with the strength of core SFRC. In this study, with the concrete strength increased by 33.7% between 38.3 and 51.2 MPa, the energy dissipation capacity increased by 32.4% between the columns A-60-CF60/1.2 and A-60-CF70/1.2.

## 4. Prediction of Bearing Capacity and Ductility

### 4.1. Bearing Capacity

The bearing capacity $N_{u}$ was taken as the peak-load in the axial load-deformation curve. The tested bearing capacity $N_{u,t}$ and the corresponding axial deformation $\delta_{u}$ of the columns are presented in Table 3. As shown in Figure 9, the bearing capacity of the columns increases linearly with the concrete strength.

The bearing capacity of rectangular-sectional CFST columns with stiffeners consists of two parts, one relates to the core concrete and steel tube, another relates to the stiffeners. In the first part, the strength of core concrete and its area are the main factors, and the effect of the strength and cross-sectional area of steel tube can be reflected in the confinement coefficient *θ*. In the second part, the yield strength and the areas of stiffeners are the main parameters. Predictive formulas of bearing capacity can be built by using the regression analysis as specified in China code GB 50936 [53] and Tao [58].
(2)$$N_{u}=f_{\mathrm{sc}}A_{sc}+f_{ys}A_{ss},$$
(3)$$f_{sc}=f_{c}\left(0.07\theta +1.92\right),$$
where, *f_sc_* is the nominal average strength of the rectangular-sectional steel tube filled with concrete; *A_sc_* is the sum of *A_st_* and *A*_c_; *A_ss_* is the cross-sectional area of the stiffeners, and the hole area of the perfobond rib should be subtracted when calculating *A_ss_*; *f_ys_* is the yield strength of the stiffeners.

Figure 10 illustrates the relation of (*N*_u−_*f*_ys_*A*_ss_)/(*f*_sc_*A*_sc_) for the columns to the volume fraction of steel fiber. Adding steel fibers has an increase on the bearing capacity, and tends to a slightly slow growth with the volume fraction of steel fiber.

Based on the mechanism subjected to loads, the effect of steel fiber on bearing capacity of the SFRCFST column relates to the bearing capacity of core concrete and steel tube. Considering the parameter of volume fraction of steel fiber and the linkage of Equation (2), the predictive formula of bearing capacity of the SFRCFST column can be built as follows,
(4)$$N_{u}=f_{c}A_{sc}\;\left(0.07\theta +1.92\right)\left(0.8v_{f}+1\right)+f_{ys}A_{ss},$$
where, $v_{f}$ is the volume fraction of steel fiber in the core concrete.

The tested values *N*_u,t_ and the predictive values *N*_u,c_ calculated by Equation (4) are presented in Table 3. The average value of *N*_u,t_/*N*_u,c_ is 1.01 with a standard deviation of 0.01. This declares a good agreement between the tested and predictive values, and the Equation (4) is rational for the prediction of bearing capacity of rectangular-sectional SFRCFST columns with stiffeners.

### 4.2. Ductility Index

To evaluate the ductility performance of the columns under axial load, a ductility index *DI* is introduced as [36],
(5)$$DI=\frac{\delta_{0.85}}{\delta_{u}},$$
where, *δ*_0.85_ is the axial deformation at 0.85*N_u_* in the descending segment of axial load-deformation curve; *δ_u_* is the axial deformation at *N_u_* of axial load-deformation curve.

The tested ductility indexes (*DI*)_t_ are listed in Table 3. *DI* decreased by 7.8% between the columns A-60-CF70/1.2 and A-60-CF60/1.2 with concrete strength increased by 33.7%. This is coming from the compression toughness of concrete decreased with higher strength. Meanwhile, the ductility index of SFRCFST columns was affected by the type of stiffener. Comparing the column B-0-CF80/1.2 stiffened with steel plate ribs to the column A-120-CF80/1.2 stiffened with perfobond ribs, *DI* increased by 12.2% from 1.718 to 1.927. As illustrated in Figure 11 and Table 3, for the columns A-120-CF80/1.2 and A-60-CF80/1.2 with the same volume fraction of steel fiber, *DI* increased from 1.927 to 1.951 with the decrease of hole spacing from 120 to 60 mm. This is due to the increased dowel action of core concrete across the holes for perfobond rib. The perfobond rib with much more circular holes benefits the ductility.

Relatively, *DI* was influenced by the presence of steel fibers. The CFST column A-60-C70/0 stiffened with perfobond ribs had a lower ductility index than equivalent SFRCFST column C-0-CF80/1.2 without stiffener, and the value decreased by 5.9% from 1.652 to 1.749. Compared to the CFST column without steel fiber, the SFRCFST columns with volume fractions of 0.8%, 1.2%, and 1.6% had the ductility index improved by 12.5%, 16.6–28% and 20.2%, respectively. As presented in Figure 12, the ductility index of the columns tends to increase linearly when the volume fraction of steel fiber is less than 1.2%, and there is a slow growth after that.

Considering the major factors of the confinement coefficient, the volume fraction of steel fiber, and the type of stiffener, the predictive formula of ductility index of rectangular-sectional CFST column can be built by using the regression analysis,
(6)$$I=0.71\theta +3.29{v_{f}}^{0.5}+0.43\left(d/s\right)+1.05,$$
where, *d* and *s* are the hole diameter and hole spacing of perfobond rib, respectively. When steel plate rib is used as the stiffener or no rib, *d* = 0.

The ductility indexes predicted by Equation (6) (*DI*)_c_ are listed in Table 3. The average value for (*DI*)_t_/(*DI*)_c_ is 0.999 with a standard deviation of 0.024. This declares a good agreement between tested and predictive values.

## 5. Conclusions

Based on the experimental study of this paper, conclusions can be drawn as follows: (1)The addition of steel fibers in core concrete improved the bearing capacity, energy dissipation capacity, and ductility of rectangular-sectional CFST columns under axial compression. The effect of steel fiber was not obvious when the volume fraction of steel fiber was over 0.8%. The concrete strength has a slight impact on the failure modes, but a significant effect on the energy dissipation capacity, bearing capacity and ductility;(2)Internal longitudinal stiffeners changed the failure modes of rectangular-sectional SFRCFST columns under axial compression. The perfobond ribs increased the ductility and energy dissipation capacity of SFRCFST columns, and worked best with steel tube when the hole spacing is two times the diameter;(3)Predictive formulas for bearing capacity and ductility index of rectangular-sectional SFRCFST columns are proposed. These give the good predictive results matched with the experimental data;(4)Rectangular-sectional SFRCFST column stiffened with longitudinal ribs can be used to reduce the cross-sectional dimensions of the structural members such as pier, abutment, pylon, and arch rib. Considering the replacement of reinforcing bars with internal longitudinal stiffeners, steel tube, and steel fibers, the construction cost of SFRCFST columns can be controlled at the same level as the normal concrete members.

## Figures and Tables

**Figure 1 materials-12-02716-f001:**
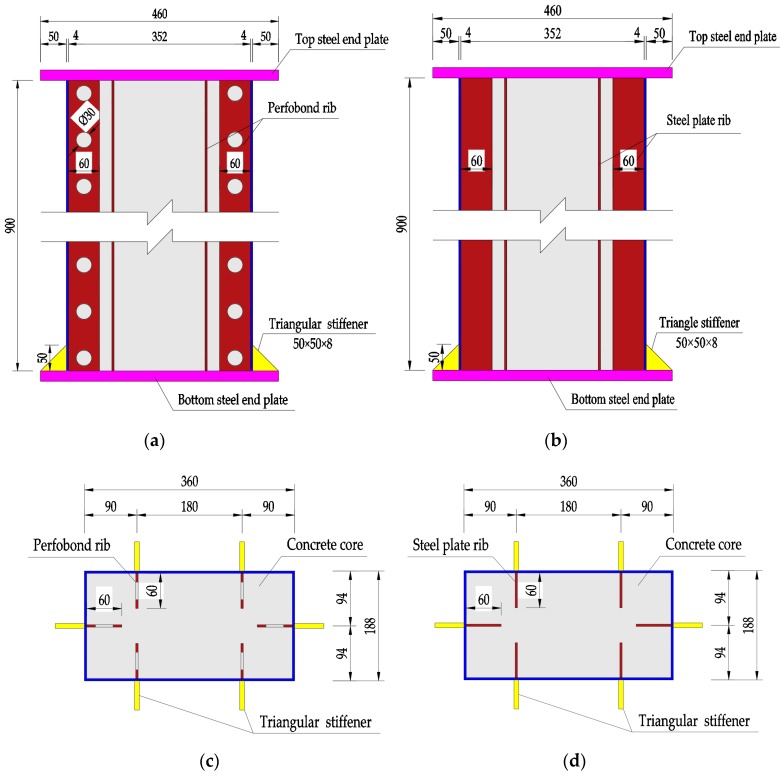
Details of columns with stiffeners (dimensions in mm). (**a**) Vertical view of perfobond ribs; (**b**) vertical view of steel plate ribs; (**c**) cross section of perfobond ribs; (**d**) cross section of steel plate ribs. Unit: mm.

**Figure 2 materials-12-02716-f002:**
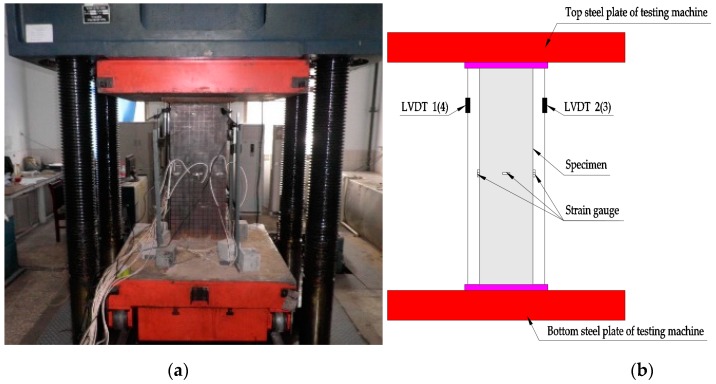
Test setup. (**a**) Photograph; (**b**) diagrammatic view.

**Figure 3 materials-12-02716-f003:**
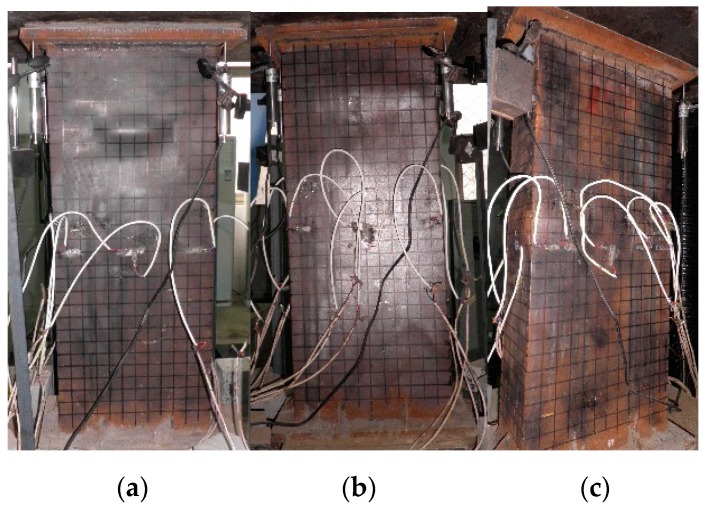
Failure modes of typical specimens. (**a**) A-60-CF70/1.2; (**b**) B-0-CF80/1.2; (**c**) C-0-CF80/1.2.

**Figure 4 materials-12-02716-f004:**
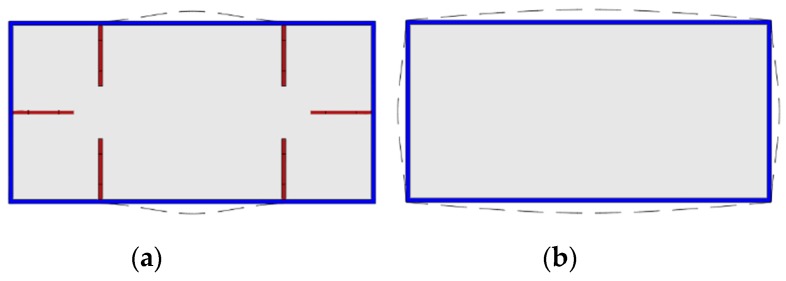
Appearances of the responding rectangular-section. (**a**) Stiffened specimen; (**b**) unstiffened specimen.

**Figure 5 materials-12-02716-f005:**
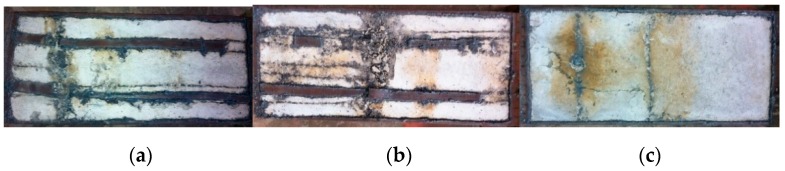
Damage degree of concrete core. (**a**) A-60-CF70/1.2; (**b**) B-0-CF80/1.2; (**c**) C-0-CF80/1.2.

**Figure 6 materials-12-02716-f006:**
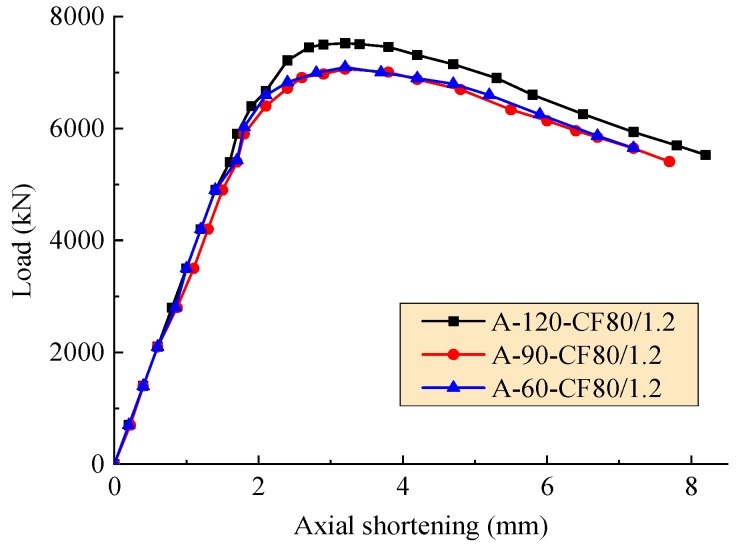
Axial load-deformation curves with varying hole spacing.

**Figure 7 materials-12-02716-f007:**
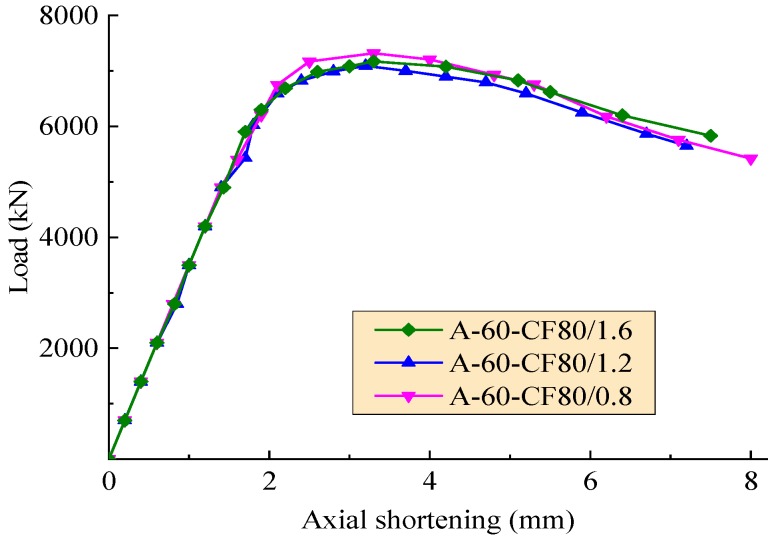
Axial load-deformation curves with a varying volume fraction of steel fiber.

**Figure 8 materials-12-02716-f008:**
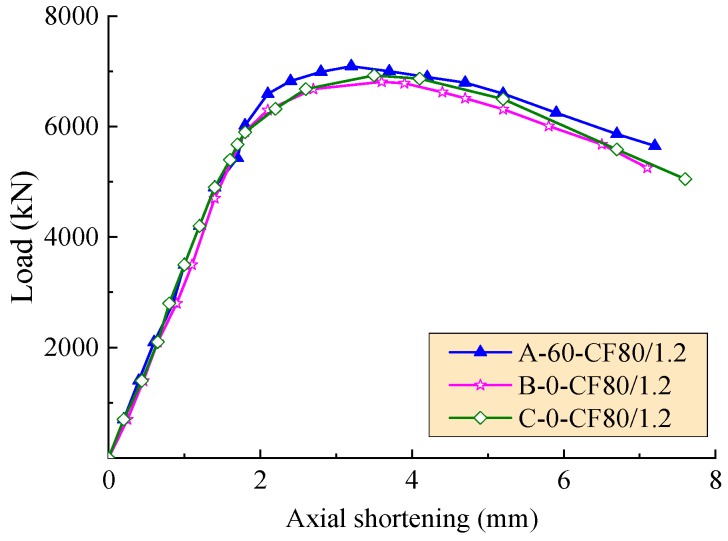
Axial load-deformation curves with a varying stiffener type.

**Figure 9 materials-12-02716-f009:**
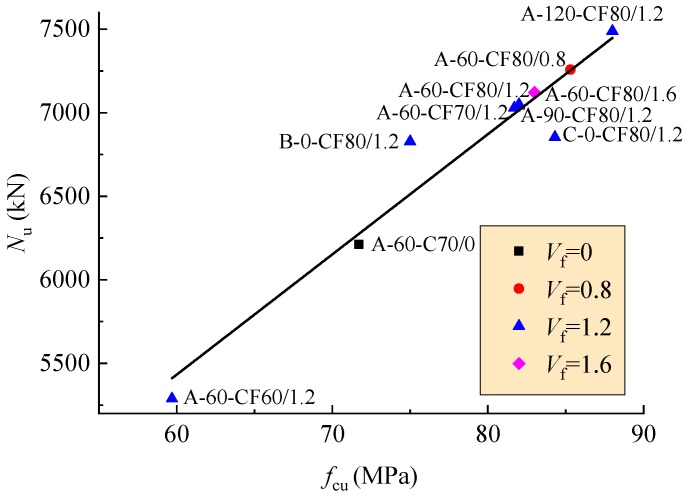
Changes of bearing capacity with concrete strength.

**Figure 10 materials-12-02716-f010:**
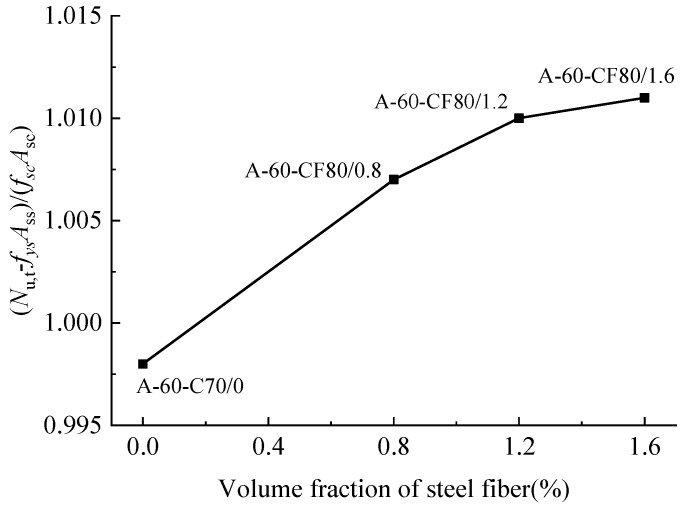
Ultimate load changed with the volume fraction of steel fiber.

**Figure 11 materials-12-02716-f011:**
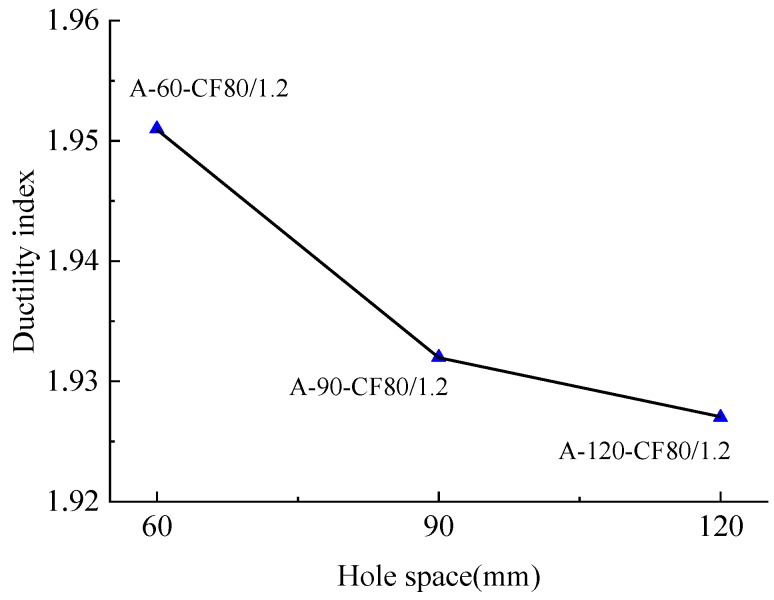
Changes of ductility index with the hole spacing of perfobond ribs.

**Figure 12 materials-12-02716-f012:**
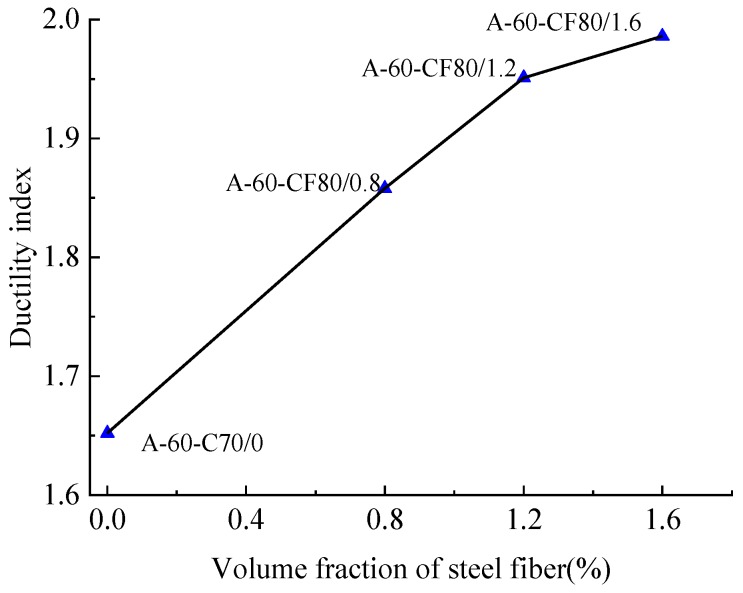
Changes of ductility index with the volume fraction of steel fiber.

**Table 1 materials-12-02716-t001:** Mix proportions of normal concrete and steel fiber reinforced concrete (SFRC).

Identifier	Cement (kg/m^3^)	Slag Powder (kg/m^3^)	Water (kg/m^3^)	Sand (kg/m^3^)	Coarse Aggregate (kg/m^3^)	Steel Fiber (kg/m^3^)	Water Reducer (kg/m^3^)
CF60/1.2	583.3	0	175	754.4	865.9	94.2	5.9
C70/0	546.6	60.7	175	605.0	1030.2	0	7.3
CF70/1.2	543.1	60.3	175	769.7	885.4	94.2	7.3
CF80/0.8	544.0	136.0	170	732.8	949.2	62.8	0.3
CF80/1.2	544.0	136.0	170	775.3	892.5	94.2	0.3
CF80/1.6	544.0	136.0	170	835.5	816.6	125.6	0.3

**Table 2 materials-12-02716-t002:** Mechanical properties and stiffener of the specimens.

Specimen ID	*f*_cu_ (MPa)	*f*_c_ (MPa)	*θ*	Stiffener Type	Hole Space(mm)	Volume Fraction of Steel Fiber (%)
A-120-CF80/1.2	88.0	54.6	0.45	Perfobond rib	120	1.2
A-90-CF80/1.2	82.0	51.3	0.48		90	1.2
A-60-CF80/1.2	82.0	51.3	0.48		60	1.2
A-60-CF80/0.8	85.3	53.2	0.47		60	0.8
A-60-CF80/1.6	83.0	51.9	0.48		60	1.6
A-60-CF60/1.2	59.7	38.3	0.65		60	1.2
A-60-CF70/1.2	81.7	51.2	0.49		60	1.2
A-60-C70/0	71.7	45.5	0.55		60	0
B-0-CF80/1.2	75.0	47.4	0.53	Steel plate rib	-	1.2
C-0-CF80/1.2	84.3	52.6	0.47	No rib	-	1.2

**Table 3 materials-12-02716-t003:** Main test results of 10 specimens.

Specimen	*N*_u,t_ (kN)	*δ*_u_ (mm)	(*DI*)_t_	*E* (MN.mm)	*N*_u,c_ (kN)	*N*_u,t_/*N*_u,c_	(*DI*)_c_	(*DI*)_t_/(*DI*)_c_
A-120-CF80/1.2	7488.3	3.1	1.927	36.77	7395.6	1.013	1.857	1.037
A-90-CF80/1.2	7040.6	3.1	1.932	34.52	6971.5	1.010	1.868	1.034
A-60-CF80/1.2	7050.6	3.2	1.951	35.65	6971.5	1.011	1.884	1.036
A-60-CF80/0.8	7257.6	3.3	1.858	35.43	7215.7	1.006	1.774	1.047
A-60-CF80/1.6	7121.7	3.3	1.986	37.67	7048.8	1.010	1.976	1.005
A-60-CF60/1.2	5288.4	2.9	2.115	26.84	5300.0	0.998	2.035	1.039
A-60-CF70/1.2	7029.1	3.2	1.951	35.53	6958.7	1.010	1.886	1.035
A-60-C70/0	6212.2	3.1	1.652	24.90	6225.3	0.998	1.669	0.990
B-0-CF80/1.2	6827.6	3.5	1.718	32.70	6663.8	1.025	1.921	0.894
C-0-CF80/1.2	6853.9	3.4	1.749	33.63	6951.9	0.986	1.863	0.939

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
