# Peer review of "Behavior of Rectangular-Sectional Steel Tubular Columns Filled with High-Strength Steel Fiber Reinforced Concrete Under Axial Compression"

_materials, 2019, doi:10.3390/ma12172716_

Round 1

Reviewer 1 Report

With respect to the English language this paper can be subdivided clearly into two parts. The abstract and the introduction are written in good English and can be understood easily. There is the term "perforbond rib" only, which would need an short explanation (see for example line 97: as comparison no stiffener). In lines 127 to 129 it is explained that the end-steel plate was welded to the steel tube. It is doubtsful if this method leads to undamaged specimens.The specimens have a cross section of 360 x 188 mm with a height of  900 mm. Specimens with these dimensions are too short to study buckling. From the text the meaning of Fig. 5 cannot be understood. From Fig 7 it is obvious that the load-deformation does not really depend on the content of steel fibers. Figure 8 shows us that different stiffeners have no significant influence on the load-axial shortening relation. This astonishing result is again due to the too short samples. Figures 10, 11 and 12 show widely scattered results, which cannot be represented with a function.

Before publication results must be critically checked and it might be difficult to come to meaningful conclusions.

Author Response

Dear Professor,

Thanks very much for your comments.

The manuscript is revised as the comments, and reply as follow:

1)With respect to the English language this paper can be subdivided clearly into two parts. The abstract and the introduction are written in good English and can be understood easily.

Response:Thanks, other parts of the manuscript have been carefully revised.

2)There is the term "perforbond rib" only, which would need an short explanation (see for example line 97: as comparison no stiffener).

Response:Ok, the explanation of the term "perforbond rib" is added in line 58-59.

3)In lines 127 to 129 it is explained that the end-steel plate was welded to the steel tube. It is doubtsful if this method leads to undamaged specimens.

Response:The end-steel plates include the top-end and the bottom-end plates. The bottom-end plate was welded with steel tube before the core concrete pumped into, the top-end was welded after the core concrete having the designed strength. Welding workers had carried out the welding process optimization before the welding construction, and welding quality and the influence on specimens were fully considered. Meanwhile, a mobile ultrasonoscope was used to do the quality inspection of the specimens before the experiment.

4)The specimens have a cross section of 360 x 188 mm with a height of 900 mm. Specimens with these dimensions are too short to study buckling.

Response:The paper presents behaviors of the SFRCFST short columns under axial compression. The slenderness ratios of the specimens, defined as the effective length divided by the radius of gyration, are 8.7 and 16.6 along the two main principal axes. They are less than the limit 17.5 of short column. The explanations are added in line 112-115. According to previous studies, the CFST short columns are prone to have “drum shape” failure mode with the confining effect being strong, while shear failure mode may happen with a weak confining effect. The local buckling of the steel tube is one of the failure modes of the short columns under axial compression.

5)From the text the meaning of Fig. 5 cannot be understood.

Response:Ok, the explanation of the Fig.5 is added in line 190-192.

6)From Fig 7 it is obvious that the load-deformation does not really depend on the content of steel fibers.

Response:The load-deformation curve was affected by the core concrete strength, volume fraction of steel fiber and type of stiffener. The former two parameters are blended in Fig 7 due to the changes of concrete strength with the volume fraction of steel fiber. The axial compressive strength of core concrete for A-60-CF80/1.6, A-60-CF80/1.2, A-60-CF80/0.8 and A-60-C70/0 were 51.9MPa, 51.3MPa, 53.2MPa and 45.5MPa, respectively. The explanation is revised, please see Line 217-225.

7)Figure 8 shows us that different stiffeners have no significant influence on the load-axial shortening relation. This astonishing result is again due to the too short samples.

Response:The blended effect of concrete strength and type of stiffener existed in Fig 8. The axial compressive strength of in-filled concrete of A-60-CF80/1.2, B-0-CF80/1.2 and C-0-CF80/1.2 were 51.3MPa, 47.4MPa and 52.6MPa, respectively. The compressive rigidity of the columns generally increased with the compressive strength of core SFRC. However, the best post-yield axial ductility took place on the SFRCFST column stiffened with perfobond rib. The larger post-yield axial ductility of the column C-0-CF80/1.2 was due to the higher compressive strength of SFRC. The explanation is revised in Line 233-237.

8)Figures 10, 11 and 12 show widely scattered results, which cannot be represented with a function.

Response:Yes, due to the blended effects of parameters studied, these figures are used to discuss the trends of relevant relationships and the three Figures have been redrawn. The functions are built based on the statistical analyses for the formulas in Section 4.

Reviewer 2 Report

The topic of the manuscript is very relevant and of interest to the structural engineering community. There a few aspects requiring addressing before it can be accepted for publication:

1 - the English writing quality is medium-low so it definitely needs to be revised and improved (preferably by a native speaker);

2 - one main question one poses when reading this study is related to cost. It is certain that the addition of steel fibers improves the performance of the filled tubes. However, does the likely high cost of the steel fibers compensate for the observed increase in performance. Other options using rubberized concrete have been recently proposed also in Europe and definitely correspond to a lower cost and should be referred together with the discussion of the aspects related to cost:

Silva, A., Jiang, Y., Castro, J.M., Silvestre, N., Monteiro, R. (2017) Monotonic and cyclic flexural behaviour of square/rectangular rubberized concrete-filled steel tubes, Journal of Constructional Steel Research, 139, 385-396. Silva, A., Jiang, Y., Castro, J.M., Silvestre, N., Monteiro, R. (2016) Experimental assessment of the flexural behaviour of circular rubberized concrete-filled steel tubes, Journal of Constructional Steel Research, 122, 557-570. Silva, A., Jiang, Y., Macedo, L., Castro, J.M., Monteiro, R., Silvestre, N. (2016) Seismic performance of composite moment-resisting frames achieved with sustainable CFST members, Frontiers of Structural and Civil Engineering, 10(3), 312-332.

3 - also related to the previous point, none of the studies included in the state-of-the-art refer to European studies, hence, additional references should be included. 

4 - in lines 46-47, the authors seem to point out only the disadvantages of using squared cross-sections but then proceed with the study on those sections. Something should be mentioned to justify the consideration of such sections anyway.

5 - in line 100, I think the authors mean respectively and not successively.

6 - in lines 176-177, when interpreting the results of Figures 6 to 8, I do not actually see a higher slope (not slop) as it is mentioned but I think they are instead rather similar. Actually, all the different options, with different percentages of fibers, do yield quite similar results. I think the comments of the results can be adapted to this. 

7 - in Equation (4) I think that parameter vf is not defined in the text.

8 - in the Conclusions, line 297, please do not say 'obviously' because if it was that obvious then the whole study would not have been necessary.

Author Response

Dear Professor,

Thanks very much for your comments.

The manuscript is revised as the comments, and reply as follow:

1)   the English writing quality is medium-low so it definitely needs to be revised and improved (preferably by a native speaker);

Response:Thanks, we have done our best to revise the manuscript carefully.

2)   one main question one poses when reading this study is related to cost. It is certain that the addition of steel fibers improves the performance of the filled tubes. However, does the likely high cost of the steel fibers compensate for the observed increase in performance. Other options using rubberized concrete have been recently proposed also in Europe and definitely correspond to a lower cost and should be referred together with the discussion of the aspects related to cost:

Silva, A., Jiang, Y., Castro, J.M., Silvestre, N., Monteiro, R. (2017) Monotonic and cyclic flexural behaviour of square/rectangular rubberized concrete-filled steel tubes, Journal of Constructional Steel Research, 139, 385-396. Silva, A., Jiang, Y., Castro, J.M., Silvestre, N., Monteiro, R. (2016) Experimental assessment of the flexural behaviour of circular rubberized concrete-filled steel tubes, Journal of Constructional Steel Research, 122, 557-570. Silva, A., Jiang, Y., Macedo, L., Castro, J.M., Monteiro, R., Silvestre, N. (2016) Seismic performance of composite moment-resisting frames achieved with sustainable CFST members, Frontiers of Structural and Civil Engineering, 10(3), 312-332.

Response:Yes, cost is one of the main factors considered in the engineering application. This paper focus on the axial compression performance of SFRCFST columns stiffened with longitudinal internal ribs, the recent research related was reviewed as Ref. [22,30-33] published at 2018 and 2017. Based on your comment, the three papers about rubberized concrete-filled steel tubes published in JCSR FSCE, 2017-2016, are added as Ref. 25-27 in the revised manuscript. The discussion of the aspects is the line 65-71.

3)   also related to the previous point, none of the studies included in the state-of-the-art refer to European studies, hence, additional references should be included. 

Response:Ok, the additional references are added as Ref. 25-29.

4)  in lines 46-47, the authors seem to point out only the disadvantages of using squared cross-sections but then proceed with the study on those sections. Something should be mentioned to justify the consideration of such sections anyway.

Response:Ok, they are revised in line 44-50.

5)   in line 100, I think the authors mean respectively and not successively.

Response:Yes, it is revised in line 124 of the manuscript.

6)  in lines 176-177, when interpreting the results of Figures 6 to 8, I do not actually see a higher slope (not slop) as it is mentioned but I think they are instead rather similar. Actually, all the different options, with different percentages of fibers, do yield quite similar results. I think the comments of the results can be adapted to this.

 Response:Yes, due to the blended effects of parameters studied, the curves did not clearly exhibit the changes of slops at ascending and descending parts. Discussion based on Figures 6 to 8 are revised, please see the revised manuscript.

7)   in Equation (4) I think that parameter vf is not defined in the text.

Response:Yes, it is added in line 291.

8)   in the Conclusions, line 297, please do not say 'obviously' because if it was that obvious then the whole study would not have been necessary.

Response:Ok, it is revised.

Reviewer 3 Report

Behavior of rectangular-sectional steel tubular columns filled with high-strength steel fiber reinforced concrete in axial compression
Shiming Liu, Xinxin Ding, Xiaoke Li, Yongjian Liu and Shunbo Zhao

The paper investigates the mechanical properties of steel tubular columns filled with high strength concrete. The investigated sections are rectangular.

The topic has interesting applications in the construction field. However, the research doesn’t appear particularly innovative and the research choices aren’t always clear and based on a scientific approach.

Indeed, it is not clear the reason of the particular realized mixes, the numerosity of the samples and the normal mixes adopted for the reference tests. It seems that there are only particular mixtures with not a clear correlation with the typical mixes and technologies used commonly in constructions.

The reviewer suggests to describe positive impact of such a research in the market and in the construction technology, with economic considerations.

Also, the reviewer recommends to furtherly comment the results because the performances of the novel materials/technology don’t show an evident and clear improvement of the behavior.

English should be improved in some parts of the paper. The reviewer recommended to improve the quality of some figures and to describe them better in the captions.

Further comments are also recommended, above all about the applicability of the results of the research, and the original contribution of the paper.

Further recommendations are reported in the attached file.

Author Response

Dear Professor,

Thanks very much for your comments.

The manuscript is revised as the comments, and reply as follow:

1)  Please modify “a referenced conventional concrete” in line 18.

Response:Ok, it is modified.

2) “its” should be replaced by “their” in line 25.

Response:Ok, it is revised.

3) The introduction is too general, please make it more tailored for the proposed research.

Response:Ok, it is substantially revised.

4)  Please add subject in line 50.

Response:Ok, it is revised.

5)  Please add space in line 50.

Response:Ok, the space is added.

6)  Please rephrase in line 63-67.

Response:Ok, they are rephrased.

7)  why rectangular sections? in line 76.

Response:It should be all the types of CFST columns and the rectangular section is deleted.

8)  Please explain why one 60 MPa, two 70MPa, three 80 MPa and also only one 70MPa normal in line 85.

Response:The number of mix proportions is designed to meet the requirement of specimens in Table.2. The explanation has been added in the revised manuscript.

9)  Please add the name of the producer.

Response:Ok, the producer is added.

10)  Not clear the numerosity of each mix design in line 97.

Response:Ok, the numerosity of each mix design is added in line 133-134 of revised manuscript.

11)  Please improve the clearity of the text in Figure. 1.

Response:Ok, the text of Figure. 1 is revised.

12) “triangle stiffener” is not clear in line 99.

Response:Ok, the explanation of the trianglular stiffener is added in line 122 of revised manuscript.

13)  Not clear the numerosity of each mix design in line 108.

Response:Ok, the numerosity of each mix design is added in line 133-134 of revised manuscript.

14)  Not clear the choice of the different mixes. Please explain.

Response:Ok, the explanation of the choice of the different mixes is added in line 129-130 of revised manuscript.

15) In what sense? “each load interval was maintained for no less than 2 min” in line 136-137.

Response:The sentence is modified to “each load interval was maintained for not less than 2 min” in line 159-160 of revised manuscript.

16) How many strain gauges? in line 140.

Response:Twelve strain gauges were used. It is added in line 167-168 of revised manuscript.

17)  Not clear. It doesn't sound scientific in line 145-148.

Response:Ok, they are rephrased in line 172-174 of revised manuscript.

18)Please add more comments in Figure 5.

Response:Ok, more comments are added in line 190-194 of revised manuscript.

19) Please quantify the gain because is doesn't seem too high in Figure 6.

Response:Ok, the explanation is revised in Line 207-212 of the revised manuscript

20) Three graphs 80 and one 70. Please modify or explain in Figure 7.

Response:Ok, the explanation is revised in Line 218-228 of the revised manuscript

21) Please explain why in line 213.

Response:The explanation is added in line 251-252.

22)  Please describe all the features present in the table 3.

Response:The explanation of the features is added in line 248,271,300-301,305-306, 337-338.

23)  Too many variables for only one regression line in Figure 9.

Response:According to the previous research and the related codes, the concrete strength is the most import parameter of the bearing capacity, and is also proved by the experimental results, so the regression line is plotted by the concrete strength.

24)  The graphs result confused in Figure 10,11,12.

Response:There are many parameters in Figures 10, 11 and 12 which show the scattered results and the three Figures have been redrawn.

As shown in Fig. 10 with the similar concrete strength and different volume fractions of steel fiber, adding steel fibers has an increase on the bearing capacity, and tends to a slightly slow growth with the volume fraction of steel fiber.

As shown in Fig. 11 with the similar concrete strength and different hole spaces, DI increased by 1.2% between 1.927 and 1.951 with the hole space decreased from 120mm to 60mm for the A-120-CF80/1.2 and A-60-CF80/1.2. This is due to the increased dowel action of core concrete across the holes for perfobond rib. The perfobond rib with a small hole space was more beneficial to the ductility index than equivalent large one.

As presented in Figure 12 with the similar concrete strength and different volume fractions of steel fiber, the ductility index of the columns increases linearly when the volume fraction of steel fiber is less than 1.2%, and there is the slow growth after that.

25)  Please rephrase line 254-257.

Response:Ok, they are rephrased.

26)  Applicability of such solutions? Which is the most performing solution?

Response:Ok, they are added in line 335-337, line44-50.

27)  about the improving with respect to normal solutions/materials? And what about costs?

Response:The discussion is added in line 347-351, and the cost is also added in line 64,347-351.

Round 2

Reviewer 1 Report

The paper has been revised and now it is certainly not totally perfect but acceptable for publication.

Author Response

Dear Professor,

Thank you so much for your compliment.

We have revised the manuscript  carefully and the corresponding revisions of manuscript were highlightened in red.

Reviewer 2 Report

The manuscript can now be accepted as is. 

Author Response

Dear Professor,

Thank you so much for your compliment.

Reviewer 3 Report

The paper investigates the mechanical properties of rectangular steel tubular columns filled with high strength concrete. The second version of the paper has been improved by the authors. However, the authors haven’t addressed all the suggestions of the reviewer, and the research work appears sufficiently investigated.

The choice of the mix design is not clear and strange comparisons among mixes with different resistance still remain. The impact of this research isn’t clear, together with the real applicability of this new technology in the constructions, under an economic point of view, too.

Further recommendations are reported in the attached file.

Author Response

Dear Professor,

Thanks very much for your comments.

The manuscript is revised as the comments, and reply as follow:

1. This description is not enough to justify the use. How much is the use of CFST columns? Why the investigation of rectangular-sectional columns is so innovative? Are they so different from square-sectional ones?

Response:Ok, the number of the related application is added in line 38-39. The description of the innovative investigation is added and revised in line 49-55 of the revised manuscript.

2. 10 specimens, 6 mix designs. Not clear the types of samples.

Response:The types of specimens including the mix designs are described in Line 131-135 in the revised manuscript. Please check it.

3. so, the load wasn't continuous? Which "load intervals" were maintained for not less than 2 minutes??

Response:Yes, the load was step-by-step applied. A load step about 7.5% of the estimated ultimate load was adopted, and each step was maintained for not less than 2 min to get the stable data. It is added in line 165-166 of the revised manuscript.

4. Not scientific. Please comment the type of failures during the test, not the sound

Response:Ok, it is revised in line 179-181 of the revised manuscript.

5. please quantify the values

Response:Ok, the calculation of slopes is added in line 211-212, and the quantify comparison is added in 216-223 of the revised manuscript.

6. Why are they compared to C70? Not comparable. Please modify.

Response:Ok, it is modified in Figure 7. The related comparison is deleted.

7. there could be peculiar behaviours related to the peculiar samples (e.g. for imperfections?)

Response:The axial compressive strength of in-filled concrete of B-0-CF80/1.2 and C-0-CF80/1.2 are 47.4MPa and 52.6MPa, respectively, and the energy dissipation capacity increases with the concrete strength. The energy dissipation capacity of B-0-CF80/1.2 has a little decrease of 2.8% because of the lower concrete strength.
